# EXcellence and PERformance in Track and Field (EXPERT)—A Mixed-Longitudinal Study on Growth, Biological Maturation, Performance, and Health in Young Athletes: Rationale, Design, and Methods (Part 1)

**DOI:** 10.3390/jfmk11010025

**Published:** 2026-01-01

**Authors:** Teresa Ribeiro, José Maia, Filipe Conceição, Adam D. G. Baxter-Jones, Eduardo Guimarães, Olga Vasconcelos, Cláudia Dias, Carla Santos, Ana Paulo, Pedro Aleixo, Pedro Pinto, Diogo Teixeira, Luís Miguel Massuça, Sara Pereira

**Affiliations:** 1Centre of Research, Education, Innovation and Intervention in Sport (CIFI2D), Faculty of Sport, University of Porto, 4200-450 Porto, Portugal; mtsr_teresa@hotmail.com (T.R.); filipe@fade.up.pt (F.C.); eguimaraes@fade.up.pt (E.G.); olgav@fade.up.pt (O.V.); cdias@fade.up.pt (C.D.); carlass@fade.up.pt (C.S.); luis.massuca@ulusofona.pt (L.M.M.); sarasp@fade.up.pt (S.P.); 2College of Kinesiology, University of Saskatchewan, Saskatoon, SK S7N 5B5, Canada; baxter.jones@usask.ca; 3Center for Sport, Physical Education, Exercise and Health, CIDEFES, Universidade Lusófona, 1749-024 Lisbon, Portugal; ana.paulo@ulusofona.pt (A.P.); pedro.aleixo@ulusofona.pt (P.A.); pedro.pinto@ulusofona.pt (P.P.); p5128@ulusofona.pt (D.T.); 4Police Research Center, Higher Institute of Police Sciences and Homeland Security, IPCOL, 1349-040 Lisbon, Portugal

**Keywords:** youth athletes, track and field, mixed-longitudinal design, biological maturation, motor performance, bioecological model

## Abstract

This paper presents the rationale and design of a study of growth and development in young track and field athletes: the EXcellence and PERformance in Track and field (EXPERT) study, and details the methodologies used. **Background**: Longitudinal research examining individual-environment interactions in youth athletic development is scarce for track and field. **Objectives**: The EXPERT study investigates how individual (anthropometry, maturation, motivation) and environmental (family, coach, club) characteristics influence developmental trajectories in youth track and field athletes. **Methods**: A mixed-longitudinal design will follow 400 athletes (200♂, 200♀; aged 10–14 years) from 40 Portuguese clubs across five cohorts assessed biannually over three years. Guided by Bronfenbrenner’s bioecological model, assessments encompass individual, performance, health, and environmental domains. Data quality control will consist of rigorous training of all research team members, implementation of standardized protocols, a pilot study, and an in-field reliability study. Multilevel growth models will examine trajectories and predictor effects of predictors. **Conclusions**: EXPERT will provide evidence to optimize training and support holistic youth athlete development.

## 1. Introduction

The development of young athletes is a complex and multifaceted process involving the interface of individual factors (e.g., physical growth, biological maturation, motor skills, cognitive development, psychosocial attributes, etc.) with environmental characteristics (e.g., familial socioeconomic status, coaching expertise, access to training infrastructure, clubs’ support mechanisms, etc.) [1,2,3]. Adolescence is recognized as a sensitive period for athletic success, as documented in the long-term athlete development (LTAD) model [4,5,6,7,8,9]. During this period, athletes progressively refine sport-specific motor skills while enhancing physical capacities in response to systematic and intensifying training and competitive demands [10,11]. Track and field exemplify a sport with diverse challenges that athletes must overcome to master various motor tasks (e.g., running, jumping, throwing). The development of such tasks can be overshadowed or masked during adolescence by physical growth, biological maturation, and psychological and social development [12,13].

Although empirical evidence regarding sports practice in pediatric populations is relatively well-documented [14,15,16], only three longitudinal studies have provided extensive information towards athlete’s guidance: from the UK, the “Training of Young Athletes” (TOYA) study [17] followed gymnastics, swimming, soccer, and tennis athletes, over 3 years; from Belgium, the “Ghent Youth Soccer Project” (GYS) [18] tracked young soccer players over 5 years; and from Portugal the “In Search of Excellence in Young Athletes” (INEX) study [19,20] monitored soccer, basketball, handball, volleyball, and water polo athletes over 3 years. Although the TOYA [17], the GYS [18], and the INEX [19,20] studies advanced understanding of youth athlete development, none examined track and field athletes. Moreover, these studies focused primarily on physical growth and motor performance, with limited consideration of psychological traits (motivation and perseverance), health indicators (nutritional status and postural control), and environmental factors (coach behaviors and club resources). These gaps highlight the need for the EXPERT study to investigate, in a broader and “theory-driven” (Bronfenbrenner bioecological system theory), the longitudinal development of young track and field athletes of both sexes. To the best of our knowledge, the Italian Athletics Federation is the only Federation that has presented results with regards to physical growth and biological maturation in young track and field athletes [21]. Additional empirical evidence regarding young athletes’ characteristics exists, but it is mostly based on cross-sectional designs [22,23,24,25], which cannot distinguish between the effects of training from those of “normal” growth and development.

Moreover, from a psychological perspective, examining motivational factors among young track and field athletes is crucial. Important indicators include perceiving sports participation as interesting and enjoyable [26,27] and how athletes persevere through obstacles, failures, and performance plateaus [28]. These factors significantly influence performance. Evidence also highlights coaches’ pivotal roles in this developmental process, as their behaviors substantially impact athletes’ achievements and psychological well-being [29]. Critical components of the coach–athlete pedagogical relationship include the nature of prescribed activities and the qualitative aspects of coach intervention/participation [30]. Additionally, the sociocultural environment in which athletes train and compete, the quality of available facilities, and familial support systems collectively contribute to athletic success [3].

To our knowledge, no systematic research project has aggregated information from young athletes, their families, coaches, and clubs into a holistic model that tracks developmental dynamics throughout adolescence using longitudinal or mixed-longitudinal methods. Contemporary young athletes, like their non-athletic counterparts, face health-related challenges from obesogenic environments that negatively impact physical activity levels, increase sedentary behavior, and affect nutritional habits. For example, global rates of overweight and obesity have increased [31,32], with 11.2% of adolescents being overweight and 6.9% having obesity [32]. Moreover, previous studies have shown declining moderate-to-vigorous physical activity [33,34,35], increased sedentary behavior [36], and reduced physical fitness levels, particularly cardiorespiratory capacity [37]. These factors potentially increase susceptibility to comorbidities, especially those of metabolic [38] and psychological nature, affecting positive/negative attitudes toward daily challenges [39].

It is hypothesized that structured sports participation during adolescence may mitigate the effects of exposure to obesogenic environments. This presents a novel challenge for coaches who must comprehensively integrate pedagogical approaches that may help them navigate the intersection between athletes’ health status, training, and competitive demands. Hence, using Urie Bronfenbrenner’s bioecological model [40] as a template, the EXcellence and PERformance in Track and Field (EXPERT) study was designed to address two main aims: (i) to follow the trajectories of physical growth, motor performance, and health-related outcomes in young track and field athletes; and (ii) to investigate the influence of individual characteristics and environmental factors on these trajectories. Complementary specific aims include: (i) modeling growth dynamics via distance- and velocity-based curves; (ii) aligning motor performance with peak height velocity to identify sensitive periods; (iii) examining trajectory associations with individual and environmental characteristics; and (iv) quantifying inter-individual variability and identifying predictors. This paper (Part 1) provides an extensive description of the EXPERT research design and its methodological approach.

## 2. Materials and Methods

### 2.1. Design

The EXPERT study was implemented in Portugal’s two largest cities—Porto and Lisbon. These cities have the highest number of affiliated young track and field athletes in Portugal (14.7% in Lisbon and 11.4% in Porto). This project was jointly designed and developed by two research centers: the Centre of Research, Education, Innovation, and Intervention in Sport (CIFI2D), in the Faculty of Sport of the University of Porto (FADEUP) and the Sports, Physical Education, Exercise and Health Research Center (CIDEFES), in the Faculty of Physical Education and Sports of the Lusófona University, Lisbon, Portugal. This research project was also conducted with the cooperative efforts of the Portuguese Athletics Federation, the Portuguese Association of Athletics Coaches, the Porto and Lisbon Athletics Associations, and their respective club members.

The Porto Athletics Association has 68 clubs spread across 17 of the 18 municipalities of the Porto region (Amarante, Baião, Felgueiras, Gondomar, Lousada, Maia, Matosinhos, Marco de Canaveses, Paços de Ferreira, Paredes, Penafiel, Porto, Póvoa de Varzim, Santo Tirso, Trofa, Valongo, and Vila Nova de Gaia). The association has 2709 registered members aged between 4 and 85 years. In the age group between 10 and 14 years, it has 683 athletes. The Lisbon Athletics Association consists of 80 clubs located across all 16 municipalities of Lisbon (Alenquer, Amadora, Arruda dos Vinhos, Azambuja, Cadaval, Cascais, Lisboa, Loures, Lourinhã, Mafra, Odivelas, Oeiras, Sintra, Sobral de Monte Agraço, Torres Vedras, and Vila Franca de Xira). The association has 3273 registered members, ranging from 3 to 89 years old. Within the 10 to 14-year-old age group, there are 629 athletes.

The EXPERT research team has developed a comprehensive set of informational materials tailored to various stakeholders involved in the study, including club directors, coaches, athletes, and parents. These materials clearly outline the objectives and rationale behind the study, providing context on why the research was being conducted and what the aims were. In addition, the materials include a summary of the expected outcomes and how these findings could benefit each group, highlighting potential advantages for athletes’ development, parents’ understanding of their children’s progress, and coaches’ strategies for training and support. Following the distribution of these materials, the research team will contact all coaches individually. These personal communications aim to establish collaborative planning for the assessment procedures, ensuring that each coach is informed and actively involved in scheduling and organizing the data collection sessions. This step is crucial for aligning the study’s logistics with each club’s training and competition routines and ensuring smooth implementation of the research protocols.

The EXPERT study adopts Bronfenbrenner’s bioecological model [40] as its conceptual template. It assumes that athlete development emerges from dynamic interactions between the subject (individual characteristics), proximal processes (e.g., coach–athlete interactions, family involvement), context (multiple environmental levels including for example, family socioeconomic and educational background, club structure and resources, availability and quality of training facilities), and time (developmental changes on performance and health occurring throughout adolescence). This perspective guided the conceptual clustering of the four study domains: individual, performance, health, and environmental (Figure 1). For more details within each domain, please see the Appendix A.

### 2.2. Sample

Athletes will be recruited from two sites—the Porto and the Lisbon Athletics Associations—aged 10–14 years at baseline. The EXPERT uses a mixed-longitudinal design with five cohorts: cohort 1, from 10 to 12.5 years; cohort 2, from 11 to 13.5 years; cohort 3, from 12 to 14.5 years; cohort 4, from 13 to 15.5 years; and cohort 5, from 14 to 16.5 years (Table 1). These ages were chosen to capture the changes in individual, performance, health, and environmental domains that occur during adolescence. Athletes will be assessed biannually every six months over three consecutive years.

The age-overlapping cohorts were designed to provide more data points across age, resulting in more precise and reliable parameter estimates in the statistical models. Furthermore, this design bears a high resemblance to the TOYA [17] and INEX [19,20] studies, and shares a similar approach with the HeartBeat project [41].

#### 2.2.1. Sample Size Calculation

The sample size calculation was determined based on conservative parameters. For cross-sectional comparisons, using a significance level of 0.05, power of 0.80, and ten groups (ages 10–14 by sex), a total sample size of 400 participants (40 per group) allows detection of small effect sizes (f = 0.20) for ANOVA: fixed effect, main effects, and interactions. For longitudinal changes, with a repeated measures ANOVA over five time points, with within-between interactions, with 10 groups (e.g., sex, age groups) will be used. A total of 400 participants (40 per group) also allows detection of small effects (f = 0.08), assuming a moderate correlation (r = 0.5) between repeated measures and a power of 0.8. If we have a dropout of 50% (~200 athletes), we can still maintain a power of 0.8 (for a two-tailed 0.05 test) and detect a small effect size of 0.12.

#### 2.2.2. Recruitment Strategy and Inclusion Criteria

All clubs affiliated with both regional track and field associations will be invited to participate in the study. Following acceptance by the clubs, the invitation will be extended to all athletes who are members of those clubs. Athletes who return signed informed consent forms from their parents or legal guardians will be included. Given this recruitment strategy, the study will rely on a convenience sample of registered young athletes engaged in regular training within development-oriented clubs. Athletes will be eligible for inclusion if they are between 10 and 14 years of age. Exclusion criteria include: (i) age < 10 or >14 years at baseline; (ii) physical, cognitive, or developmental disabilities requiring adapted training; (iii) acute or chronic conditions contraindicating physical testing (per medical documentation); and (iv) inability to provide informed consent/assent.

### 2.3. Procedures

Each assessment session will be selected according to the competitive calendar of each age category and club. The assessment sessions will be structured to maximize efficiency and accuracy, covering either a morning period (9:00 a.m. to 12:30 p.m.) or an afternoon session (2:00 p.m. to 5:30 p.m.). Regardless of the time of day (morning or afternoon), the sequence of assessments will remain consistent. This scheduling flexibility will allow for greater participation from clubs and athletes, ensuring optimal assessment conditions.

On assessment days, all athletes, along with their parents and coaches, will be welcomed, and the research questions again be explained. Then, all athletes will complete the questionnaires (in person) related to their practice, motivation and resilience, perception of health status and positive development, and family structure and support under the supervision of the research team. Following the completion of questionnaires, athletes will proceed with the anthropometric measurements and the dynamic postural control assessment (in distinct groups, but always in this order). Once these assessments are completed, the athletes will complete a warm-up before the motor performance-related evaluations. The general warm-up will consist of continuous running (~5 min, low intensity), joint mobility exercises (foot, hip, shoulder circles; knee flexion/extension; lunges; torso twists; leg swings), running technique drills (arm swings, ankle drills, skipping), and short accelerations (10 m). Total duration: approximately 20 min. After that, they will be divided into groups and tested for the performance of static strength, running speed, abdominal muscular strength, lower-body explosive strength, upper-body explosive strength, and aerobic component (Figure 2).

All assessments will occur between April/May and November/December, with a 20 to 30-day window to allow for athletes’ training regimens and competitive schedules. Informed consent forms will be given to the coaches, who will distribute them to the parents or legal guardians. Only athletes with signed consent forms will be included, ensuring proper assent procedures are followed for minors. The study was approved by the Ethics Committee of the lead institution (CEFADE 38.2023).

### 2.4. Measurements

Following the EXPERT design, measurements are clustered into four domains: (i) individual, (ii) performance, (iii) health, and (iv) environmental.

#### 2.4.1. Individual Domain

##### Physical Growth—Anthropometry and Body Composition

Height, sitting height, arm length, leg length, and waist circumference will be measured. Height and sitting height will be measured using a stadiometer (Seca 213 stadiometer, Seca GmbH & Co. KG, Hamburg, Germany). With the athletes in the standard anthropometric position and aligned heads to the Frankfurt plane, measurements will be taken with a precision of 0.1 cm. For sitting height, athletes will be seated on a 40-cm-high bench with their hands on their legs. Arm length will be measured with a Rosscraft anthropometric segmometer (Rosscraft, Rosscraft Innovations Inc., Surrey, BS, Canada), while leg length will be estimated as the difference between height and sitting height. Waist circumference will be measured using a non-elastic tape measure (Rosscraft Anthro tape, Rosscraft Innovations Inc., Surrey, BS, Canada). All measurements will be taken twice. For height, a third measurement will be obtained if a difference greater than 3 mm occurs, and a difference of 2 mm will be considered in the remaining measurements. The average of the two nearest measurements will be used for analysis. All measurements will follow the standards of the International Working Group on Kinanthropometry protocols [42].

Body mass, body fat percentage, and fat-free mass will be assessed using a portable bioimpedance scale (Tanita BC-553, Tanita Corporation, Tokyo, Japan). Athletes step onto the Tanita barefoot, without socks, and wearing shorts (and a top for girls). They will stand still in a stationary position. Body mass index (BMI) will be calculated using the standard formula [weight (kg)/height (m^2^)].

##### Biological Maturation

The biological maturation will be assessed using the maturity offset methodology described by Mirwald, Baxter-Jones, Bailey, and Beunen [43], which estimates the temporal distance, measured in years, relative to the age at which an individual experiences peak height velocity (PHV). The maturity offset is expressed as a positive (+) or negative (−) value. A positive maturity offset signifies that the individual has already attained their PHV, and a negative (−) value means that the individual has yet to reach their PHV.

Although the maturity offset method has recognized limitations (prediction error ≈±1 year, reduced accuracy in early/late maturers), it remains the most feasible non-invasive approach for large-scale field studies. Sensitivity analyses will compare results using chronological age versus maturity offset to assess the impact of estimation error [44].

##### Motivation

The intensity of the athletes’ intrinsic motivation will be assessed with the Portuguese validated version of the Intrinsic Motivation Inventory [45] by Fonseca and Brito [46]. This version includes four dimensions: interest-enjoyment, perceived competence, effort, and pressure-tension. In addition, the relatedness and perceived choice subscales originally included in McAuley et al.’s [45] version will be incorporated into the present study. These subscales are translated and specifically adapted for this study to ensure linguistic and conceptual equivalence within the target population. Responses will be recorded on a 5-point Likert scale, ranging from 1 (I totally disagree) to 5 (I totally agree).

##### Perseverance

The Short Grit Scale (Grit-S) [47] assesses perseverance and sustained interest in long-term goals. The original version consists of eight items distributed across two subscales: perseverance of effort (4 items), which reflects an individual’s ability to maintain effort over time to achieve a specific goal, and consistency of interests (4 items), which relates to the stability of one’s long-term commitment to a given task. In the validated Portuguese version [48] to be used in this study, one item from the perseverance of effort subscale was removed, resulting in a final structure of seven items. Athletes will respond using a 5-point Likert scale, ranging from 1 (strongly disagree) to 5 (strongly agree).

The translated/adapted items have been psychometrically validated in Portuguese youth populations. The Portuguese version of the Intrinsic Motivation Inventory [46] demonstrated adequate internal consistency (Cronbach’s α: 0.70–0.85) and satisfactory four-factor structure fit. The Portuguese Short Grit Scale (Grit-S) [48] showed acceptable reliability (perseverance: α = 0.78; consistency: α = 0.73) and model fit (CFI = 0.94, RMSEA = 0.06), supporting construct validity in this population.

#### 2.4.2. Performance Domain

##### Motor Performance

Static strength

Static strength will be assessed using the handgrip strength test. Athletes will stand without contacting any surface or wall. The grip bar of the hand-held dynamometer (Takei Digital Grip Strength Dynamometer Model T.K.K.5401, Takei Scientific Instruments, Tokyo, Japan) will be adjusted to fit comfortably in their hands. With the arm extended and away from any part of the body, athletes will squeeze gradually and continuously for at least 5 s, performing maximal strength (kgf). They will repeat this procedure with the other hand. The best performance of both hands in two trials will be used as the test result [49].

Running speed

Running speed will be measured using the 40 m test. Time will be recorded using the photoelectric cell system Speed Trap II (Brower Timing Systems LLC., Draper, UT, USA), with pairs of cells placed at 5, 10, 20, 30, and 40 m splits. Split times will be collected to characterize performance across acceleration phases (0–5 m: reaction/initial acceleration; 5–20 m: acceleration; 20–40 m: maximum velocity transition) rather than to derive mechanical force-velocity parameters. Athletes will take their starting position behind the 0-m line and, following the commands “on your marks,” “ready,” and “go,” will run in a straight line at maximum velocity. Each athlete will perform two trials, and the best result will be recorded [50].

Abdominal muscular strength

Abdominal muscular strength/endurance will be assessed using the sit-up test. Athletes will perform full sit-ups with their knees flexed at an angle less than 90°. Their feet will be held on the floor, and both hands will be placed behind their head. A helper will secure their feet and will count one repetition each time the athlete’s elbows touch their knees. Athletes will complete the maximum number of repetitions in 60 s. Only one trial will be performed [51].

Upper-body explosive strength

Upper-body explosive strength will be assessed with the seated medicine ball throw test [52] and the seated single-arm shot put throw test [53] for both hands/sides. For the seated medicine ball throw test, athletes will sit on the floor with their back and shoulders against a wall or support, legs together and extended. Holding the medicine ball (Topgim, 3 kg, Lisboa, Portugal) close to their chest with both hands and elbows flexed, athletes will throw it horizontally as far as possible while maintaining contact with the wall. For the seated single-arm shot put throw test, athletes will maintain the same position as in the first test but will push the medicine ball with only one hand. The ball will be positioned next to the throwing shoulder, with the opposite hand holding it. Athletes will also be instructed to throw the medicine ball as far as possible in the horizontal direction without losing contact with the wall. In both tests, the throw will be measured from the wall (zero measurement line) to the landing point. The athletes will perform three trials for each test, and their best trial (distance in meters) will be used as the test result.

Lower-body explosive strength

Four tests, clustered into horizontal and vertical procedures, will be used to assess lower-body explosive strength. For the horizontal procedure, athletes will perform a standing long jump [54]. The athlete will stand with feet parallel behind a starting line. After the signal, they will swing their arms and will jump with both feet simultaneously as far forward as possible. The jumps will be measured from the starting point to the heel of the closest foot. Athletes will perform three trials; the best trial will be used as the test result. The squat jump, countermovement jump, and 10/5 repeated jump test will be used for the vertical procedures. Athletes will perform the squat jump and the countermovement jump, as described by Bosco et al. [55]. For the squat jump, the athlete will start standing with feet parallel and shoulder-width distance. After the command “ready,” the athlete will position themselves in a squat close to a 90° angle. At the command “go”, the athlete will jump as high as possible. In the countermovement jump, the athlete will also start in a standing position and, at the signal “go,” will jump as high as possible after a fast flexion of the hips and knees. Each athlete will perform three trials of each test (with 30 s of rest between jumps) and will register the best jump height as the result.

The 10/5 repeated jump test (RJT) will be used to measure the lower-body reactive strength index (RSI). Athletes will perform one trial [56], as described by Harper et al. [57]. They will be instructed to perform optimal vertical rebounds, with maximal elevation and minimal ground contact at each jump, keeping their hands on their hips. The athletes will perform eleven jumps, but the first one will be discarded since it does not involve the fast stretch-shortening cycle. The RSI score will be obtained by the average of the best 5 jumps with the highest RSI, and this score will be used as the result of the RSI.

All the vertical jumps will be performed in the Chronojump Boscosystem mat (Chronojump System, Barcelona, Spain).

##### Aerobic Component

To assess the aerobic component, the 1000 m test will be used [58]. Athletes will have to complete the distance by running and walking, if necessary, in the shortest time possible. The test will be on a homologate track (400 m per lap), where athletes will have to complete two and a half laps. They will begin in the starting position on the appropriate line and will start after the signal (“on your marks” and “go”). This test will be performed once, with results recorded in seconds (Geonaute Watch Chrono 700, Shanghai, China).

The selection of motor performance tests was based on principles from available long-term athlete development models (LTAD [4], PYD [5], FTEM [6], and CYD [7]), which recommend prioritizing general and transferable physical capacities during the early stages of development rather than sport-specific technical skills. Accordingly, standardized motor performance tests that assess fundamental qualities relevant across all track and field disciplines will be used, as they are suitable for athletes with varying levels of experience. Event-specific technical tests will be excluded because they vary significantly across disciplines, depend on athletes’ prior experience, and offer limited comparability. The approach used ensures a fair, consistent, and developmentally aligned evaluation of athletes’ motor performance.

##### Competition Outcomes

The competition performance results will be systematically collected and analyzed via the comprehensive official competition results gathered during the present sports season. Competition results will be registered for each athlete, selecting the best result achieved (time and distance) across various athletic events. These events will encompass several disciplines, including sprint events, long jump, high jump, shot put, and endurance events. All performance data will be sourced directly from the detailed rankings databases of the partner associations involved in this project, provided by their respective technical directors.

#### 2.4.3. Health Domain

##### Nutritional Status

Nutritional status will be based on BMI (body mass index) using cutoff values provided by the World Health Organization [59] to categorize athletes as underweight, normal weight, overweight, or obese. Furthermore, using cutoff values from McCarthy et al. [60] to body fat—the 85th and 95th percentiles for boys and girls adjusted for their age, will be used to define over-fat and obesity. Waist circumference as a marker of abdominal obesity will also be used based on cutoffs provided by Cook et al. [61] to define abdominal obesity (≥ 90th percentile).

##### Aerobic Component

To estimate the aerobic component, the 1000 m run will be assessed, as described in the previous section, using the completion time as a result. VO_2_max will also be estimated using the formula suggested by Klissouras [62]. The absolute and relative frequency of VO_2_max will be classified according to Matsudo’s Reference Table [63].

##### Dynamic Postural Control

Dynamic postural control will be assessed with the Hurdle Step test [64] in a stratified random subsample (55%, n ≈ 220), given time constraints. This subsample provides adequate power (>0.80) to detect medium effects (f = 0.25) while reducing assessment burden. Athletes will be randomly selected within sex × cohort strata at baseline and reassessed at all time points. In addition to assessing postural stability, the collected information will allow detection of potential bilateral mobility and joint stability issues in the hips, knees, and ankles. The test will also challenge the stability and control of the pelvis and core while allowing one to observe functional symmetry. The athletes will perform five repetitions with each leg, and all repetitions will be scored on a categorical scale from 0 to 3, according to the FMS guidelines.

##### Perception of Health Status and Positive Development

Health status

The Health Behavior in School-aged Children (HBSC) questionnaire [65] will be used to assess perceived health status, quality of life, well-being, and emotional state. Perceived health status and well-being will each be measured with a single item rated on a 4-point Likert-type scale ranging from 1 (poor) to 4 (excellent). Quality of life will be assessed using 10 items rated on a 5-point Likert-type scale from 1 (never) to 5 (always). Current emotional state will be evaluated with five items assessing happiness, calmness, energy levels, sleep quality, and engagement in meaningful activities, each rated on a 6-point Likert-type scale from 1 (never) to 6 (all the time). This multidimensional approach will comprehensively assess adolescents’ self-perceived health and well-being.

##### Positive Development

Positive development will be evaluated using the Personal and Social Responsibility Questionnaire (PSRQ), originally developed by Watson et al. [66] and later modified by Li et al. [67]. The Portuguese version, validated by Martins et al. [68], comprises 14 items divided into two dimensions: personal responsibility and social responsibility. Each item is rated on a 6-point Likert-type scale, ranging from 1 (strongly disagree) to 6 (strongly agree).

#### 2.4.4. Environmental Domain

##### Family Structure and Support

All parents or legal guardians will complete a questionnaire that will gather information on demographics (e.g., sex, age, education level), household size, and lifestyle factors. Furthermore, a questionnaire developed by Fredricks and Eccles [69] and cross-validated for the Portuguese population by Dias and Fonseca [70] will be used to assess parental support and encouragement for sports participation. It will include items covering four domains related to each parent’s perceptions: (i) the child’s sporting ability, (ii) the value of sport, (iii) the level of support and encouragement provided, and (iv) the time spent with the child in sport and purchasing sports equipment. Answers will be coded on a 5-point Likert scale: in the (i) child’s sporting ability, it will range from 1 (very poor) to 5 (very good); in the (ii) value of sport from 1 (no importance) to 5 (very important); in the level of (iii) support and encouragement provided from 1 (strongly discourage) to 5 (strongly encourage); in the (iv) purchasing sports equipment from 1 (rarely) to 5 (frequently). To assess how much time parents report spending with the child on sports activities over the past week, one item (hours/week) will be used.

##### Coach’s Knowledge and Competence

Coaches will be requested to answer an online questionnaire concerning demographics (age and gender) and sports involvement (academic and sports education and professional experience). Furthermore, to assess coaches’ knowledge and competence, the Coaching Efficacy Scale developed by Feltz et al. [71], cross-culturally validated for the Portuguese population by Duarte et al. [72], will be used, covering four dimensions: (i) planning strategy, (ii) motivation, (iii) teaching technique, and (iv) character building. Answers will be provided on a 5-point Likert scale ranging from 1 (not important) to 5 (totally important). The Leadership Scale for Sports Scale developed by Chelladurai and Saleh [73], cross-culturally validated for the Portuguese population by Cruz and Chelladurai [74], will also be used, covering five domains: (i) training and instruction, (ii) democratic behavior, (iii) autocratic behavior, (iv) social support, and (v) positive feedback. All items in the two questionnaires will employ a 5-point Likert scale, ranging from 1 (never) to 5 (always). Finally, information will also be obtained regarding coach intervention, namely practice conditions and augmented feedback, with respect to age groups and skills practiced (runs, jumps, and throws). Concerning practice conditions, the following will be collected: (i) practice schedule; (ii) part versus hole practice—in partial (segmentation, simplification, and/or fractionation); and (iii) practice variability—multiple skills versus one-skill sessions, and in a single skill, constant versus varied practice [75]. Regarding augmented feedback, the focus will be on: (i) who? (individual or group); frequency and possible strategies used—summary feedback, bandwidth feedback, self-determined feedback; (ii) timing (instruction, during/after execution); (iii) content—attentional focus (internal or external), knowledge of results or knowledge of performance; (iv) form—verbal (prescriptive, questioning, positive or negative), modeling (coach/peer/other), video (self/other, alone or with the coach, intervening or not, and if so, how), biofeedback, and physical assistance [75]. To the best of our knowledge, this study will be the first to aim for an overview of track and field coaches’ intervention. An exploratory and descriptive approach will be followed, using a survey questionnaire in which responses will be given on a presence/absence or ratio (% of total) basis.

##### Club Information

Using a questionnaire developed by the research team, a set of questions will be organized into four domains: (i) club characterization—location; number of sports; number of athletes in track and field; club structure (events and competitive levels); and years since the foundation; (ii) club infrastructure—practices location; characterization of facilities; complementary rooms/places (facilities); (iii) human resources—characterization of staff; number of coaches; coaches’ level category certification; and (iv) club communication—communication manager; social media; tv/online channel and/or radio station; promotion of competitions and events. A member of the club will answer the questionnaire.

### 2.5. Data Quality Control

Several procedures will be carried out to ensure data quality control. First, all research team members will participate in a comprehensive training program that includes instruction on all protocols, supervised practice, and formal certification before being allowed to conduct assessments. To support long-term standardization, a detailed manual of procedures has been developed and will be provided to all researchers, outlining each test and measurement. Intensive training sessions will be conducted, and supplemental training will be provided as needed to ensure the collection of high-quality data throughout the study. Second, a pilot study will be conducted with athletes from one club prior to the main data collection. This hands-on testing will serve to: (i) evaluate and refine all protocols in real conditions; (ii) assess data collection, preparation, and organization procedures; (iii) test the adequacy of instruments, recruitment strategies, and time allocation per domain; and (iv) identify potential methodological issues. Insights and feedback gathered during this preliminary phase will inform a series of targeted adjustments, thereby enhancing the overall reliability and validity of the main study. Third, a random sample of athletes will be re-evaluated throughout data collection to ensure in-field reliability, and adjustments will be made whenever needed to ensure reliability. For the FMS Hurdle Step, two experienced assessors will be responsible for all evaluations in both Porto and Lisbon. Although test–retest procedures will not be conducted due to the protocol’s length and the risk of athlete fatigue, the use of the same trained assessors in each city will help maintain consistency in scoring. Fourth, data entry will be verified, and cleaning procedures will be applied to ensure highly accurate data for statistical analysis. Finally, throughout the entire data collection period, the same assessment locations, equipment models, and evaluators will be consistently maintained, thereby minimizing variability in testing conditions. All instruments will be calibrated prior to each session, in accordance with the manufacturer’s guidelines.

### 2.6. Statistical Analysis Plan

Based on the specific aims of the EXPERT study, data analysis is outlined to address each aim.

First, as is commonly done in all analyses, an exploration analysis will be conducted to detect potential outliers and examine the distribution of all variables. Additionally, missing data patterns will be analyzed within each cohort to determine their nature, i.e., missing at random or missing completely at random. Sensitivity analyses will be conducted to assess the robustness of conclusions under different analytical assumptions. Cohort effects in the analysis will be controlled by introducing dummy codes.

For aim 1 (modeling the dynamics of physical growth and constructing distance- and velocity-based growth curves), the Bock, du Toit, and Thissen triple logistic model will be used using the AUXAL software (version 3.1).

Age at peak height velocity (PHV) will serve as a milestone for analyzing the timing of growth during puberty.

For aim 2 (to align motor performance with the age at peak height velocity and to identify potential sensitive periods in the development of motor components), time will be conceptualized in terms of biological age (e.g., age from PHV). Peak spurts in height and motor performance will be identified. Sensitive periods will be defined as the period between maximum gain increase before the peak spurt and maximum loss after. Mean velocity curves will be developed in biological age (months before/after PHV) based on two criteria: (i) low measurement error and (ii) a single velocity increase.

For aim 3 (describing trajectories in motor performance and health indicators, and their associations with individual growth, biological maturation and psychological traits) and environmental (coach, club and family) characteristics, multilevel growth models will be used to describe longitudinal trajectories of motor performance and health indicators, accounting for the hierarchical structure of the data (e.g., repeated measures nested within individuals, which are nested within clubs).

For aim 4 (investigating the magnitude of inter-individual variability in intra-individual trajectories of motor performance and health indicators, and identifying their predictors), multilevel growth models with random effects will be applied to estimate inter-individual variability in intra-individual trajectories. Variability will be quantified by decomposing variance at different levels (e.g., variance between individuals and variance within individuals). These models will also allow for the identification of predictors of inter-individual variability, including individual characteristics (e.g., growth, biological maturation, psychological traits) and environmental factors (e.g., coach, club, family).

## 3. Discussion

It is well understood that various individual and environmental factors influence the development processes of young athletes [76]. Although this “tapestry” has been described in both team sports [19,20] and individual sports [17], and framed within several long-term athlete developmental models [4,5,6,7], empirical research simultaneously examining these factors within a developmental framework remains limited, particularly from a bioecological perspective [40].

In track and field, despite the existence of youth-oriented programs in several countries [77,78,79,80,81] that tackle this issue in the context of the idea of “kids athletics”, available data are cross-sectional and lack evidence-based support linking individual characteristics [22,23,24,25] and sporting environment [3]. Understanding how young athletes develop requires longitudinal data to reveal the multifaceted relationships between factors that shape their performance trajectories. In track and field, longitudinal studies are scarce and have largely focused on competitive performance outcomes [82,83,84,85]. Despite the valuable contributions of these reports, none appear to have addressed the intricate complexities of individual and/or environmental factors. To gain a broader understanding of athletic development, it is essential to also consider these factors.

Grounded in the bioecological perspective [40] and informed by long-term athlete development models [4,5,6,7,8,9], the EXPERT study aims to advance current knowledge in several important ways. First, it focuses on athletes aged 10 to 16 years, an essential time window for growth and motor development, i.e., adolescence. Second, it examines a wide range of predictive factors, encompassing both individual athlete characteristics and environmental information from parents, coaches, and clubs. Third, the use of a mixed-longitudinal design—combining multiple age cohorts with overlapping assessment periods— allows for more rapid data collection. Four, the project uses Bronfenbrenner’s [40] bioecological theory as a template, which emphasizes the dynamic interplay between individuals and the multiple environmental systems in which they are embedded.

It is also important to name some of the EXPERT limitations. First, although the Track and Field Associations have 26.1% of the affiliated young Portuguese athletes, the project was implemented in only two sites—Porto and Lisbon—this may not reflect the diversity of sports contexts across the country. Likewise, the participating clubs may not represent the conditions of clubs in other regions, particularly those with fewer resources. Second, it focuses on athletes aged 10 to 16, capturing a specific phase of development, but excluding others. Third, specific technical skills in track and field were not assessed. Fourth, as a mixed-longitudinal study, there is also potential for participant dropout over time. Fifth, the EXPERT study uses a convenience sample where participant dropout is possible, and no causal inferences can be made. Consequently, the results may not be generalizable to all young Portuguese track and field athletes.

Irrespective of these limitations, the EXPERT also has strong points, and the following main outcomes are expected: (1) at the athlete level, the goal is to obtain a comprehensive, extensive, and integrated data set of athlete growth and development, as well as to identify windows of opportunity for achieving excellence across various events by optimizing responses to training. Additionally, relevant information on the athletes’ health status will be gathered to support safe and sustainable development; (2) at the parent level, the aim is to obtain relevant data regarding the importance of encouragement and support provided to their children in achieving high-level sports performance throughout their athletic careers; (3) at the coach and club level, the data will provide evidence to improve the quality of training methodologies—enhancing the planning and structure of training sessions by considering sex differences and maturational variability—as well as to refine athletic development programs aimed at promoting more sustainable long-term development. Moreover, the EXPERT study can help identify sensitive periods in the development of motor performance, which can be utilized by coaches to plan and design their training regimens. Additionally, understanding the interaction between performance domains can help prioritize development goals within clubs, regional associations, and the national track and field federation, ultimately supporting long-term athletic success; (4) at the level of local sports associations and the national federation, the data may support the development of evidence-based sports policies, such as the creation of a more integrated and effective program focused on youth athletic development. The results can provide valuable evidence to guide long-term athlete development strategies and inform decision-making in youth sports structures; and (5) at the research level, the generation of innovative data is expected to contribute to the production of scientific publications.

## 4. Conclusions

Long-term athlete development is a dynamic and multifaceted process that requires an integrated approach, considering both individual characteristics and environmental factors. The EXPERT study aims to enhance the understanding of the dynamic network between individual characteristics and environmental factors, as well as the influence of the environment on the physical growth and developmental performance of young track and field athletes as they progress throughout their sporting careers.

## Figures and Tables

**Figure 1 jfmk-11-00025-f001:**
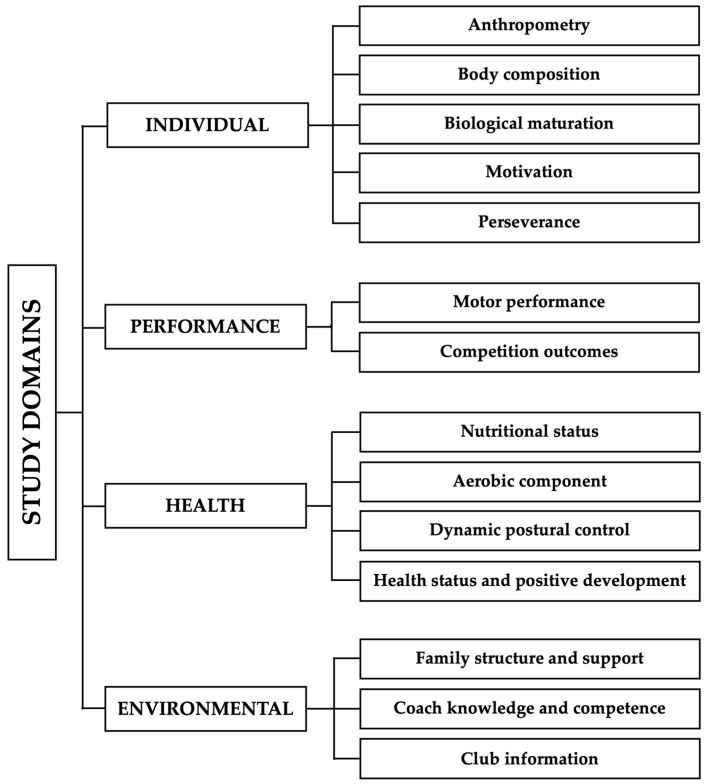
A summary overview of the four domains of the EXPERT study: individual, performance, health, and environmental. Each domain contains specific information that will be assessed in young track and field athletes.

**Figure 2 jfmk-11-00025-f002:**
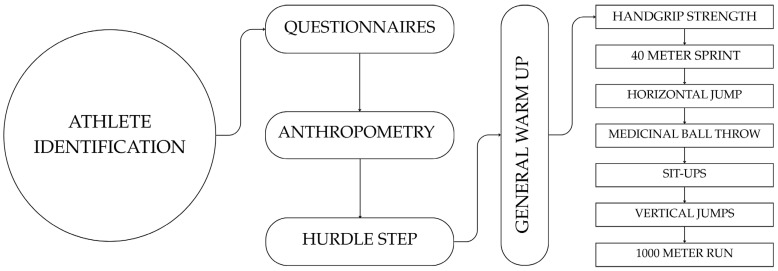
Assessment sequence in the EXPERT study. Following athlete identification and questionnaire completion, participants undergo anthropometry and hurdle step assessment. After a standardized warm-up, athletes complete motor performance tests in the following order: handgrip strength, 40 m sprint (with splits), sit-ups, medicine ball throws, jumps (standing long jump, squat jump, countermovement jump, reactive strength), and a 1000 m run.

**Table 1 jfmk-11-00025-t001:** Suggested distribution of participants across cohorts, overlapping ages, and total sample size per cohort. Each cohort represents a group of participants followed over a specific age range (ages in bold indicate the starting age of each cohort). Columns show participants’ suggested ages at assessment, the number of females (♀) and males (♂) in each cohort, and the total per cohort. The bottom row displays the total number of participants across all cohorts, by individual sex, and overall total.

Cohorts	Age (Years)														♀	♂	Total
Cohort 1	**10.0**	10.5	11.0	11.5	12.0	12.5									40	40	80
Cohort 2			**11.0**	11.5	12.0	12.5	13.0	13.5							40	40	80
Cohort 3					**12.0**	12.5	13.0	13.5	14.0	14.5					40	40	80
Cohort 4							**13.0**	13.5	14.0	14.5	15.0	15.5			40	40	80
Cohort 5									**14.0**	14.5	15.0	15.5	16.0	16.5	40	40	80
Total															200	200	400

## Data Availability

The data presented in this study are available upon request from the corresponding author.

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
