# Peer review of "EXcellence and PERformance in Track and Field (EXPERT)—A Mixed-Longitudinal Study on Growth, Biological Maturation, Performance, and Health in Young Athletes: Rationale, Design, and Methods (Part 1)"

_jfmk, 2026, doi:10.3390/jfmk11010025_

Round 1
Reviewer 1 Report
Comments and Suggestions for Authors
Peer Review Report
Title: EXcellence and PERformance in Track and Field (EXPERT) — A mixed-
longitudinal study on growth, biological maturation, performance, and health in young
athletes: Rationale, design, and methods (Part 1)
1. General Comments
The manuscript presents the rationale, design, and methodological framework of the
EXPERT study, an ambitious mixed-longitudinal project addressing growth, maturation,
performance, health, and environmental determinants in young track and field athletes.
The study fills a clear gap in the current literature, where comprehensive longitudinal data
in this sport are scarce. The rationale is strong, the methodological design is detailed, and
the authors frame the project within the bioecological model, which adds conceptual
rigor.
Overall, the manuscript is well structured, readable, and grounded in relevant literature.
Its main strength lies in the breadth of variables considered across multiple ecological
domains. However, the manuscript would benefit from clearer articulation of specific
research questions, improved coherence between the theoretical framing and the
operationalization of measures, and more precision in describing certain methodological
procedures. Some sections are verbose and may benefit from stylistic tightening to
improve clarity.
2. Major Issues
1. Clarity of Research Questions and Hypotheses
While the rationale is comprehensive, the manuscript does not articulate explicit research
questions or testable hypotheses. A large-scale longitudinal design such as this one should
clearly define:
• which developmental trajectories are expected,
• how specific individual and environmental predictors are hypothesized to operate,
• and the anticipated interactions across domains.
Recommendation: Include a concise subsection outlining primary and secondary
research aims or hypotheses to enhance conceptual clarity and guide future analyses.
2. Integration of the Bioecological Framework
Although Bronfenbrenner’s bioecological model is invoked as the guiding framework,
this theoretical lens is not consistently integrated with the measurement plan. The four
measurement domains correspond to ecological levels, but the manuscript does not
explicitly map them onto microsystem, mesosystem, exosystem, and macrosystem
processes. Without this mapping, the theoretical connection remains somewhat
superficial.Recommendation: Strengthen the theoretical-methodological integration by explicitly
indicating how each measured variable operationalizes components of the model and how
the design will allow examination of person–process–context–time interactions.
3. Methodological Transparency in Sampling and
Recruitment
The manuscript states that 400 athletes will be recruited, but it is unclear:
• how clubs were selected,
• how representativeness is ensured,
• and what inclusion/exclusion criteria apply (e.g., training experience, injury
status).
Given the heterogeneity across clubs and cities, more detail is needed to assess potential
biases.
Recommendation: Add a clear description of inclusion criteria, recruitment strategies,
and any procedures to minimize sampling bias.
4. Measurement Reliability and Standardization
Although the manuscript mentions rigorous training and pilot studies, reliability
procedures for several tests (e.g., jumps, dynamometry, questionnaires) lack detail.
Mixed-longitudinal designs rely heavily on measurement consistency across years,
locations, and assessors.
Recommendation: Specify inter-rater and intra-rater reliability protocols and anticipated
reliability thresholds, especially for subjective measures such as the FMS Hurdle Step or
questionnaire-based constructs.
5. Statistical Analysis Plan
The manuscript provides sample size justification but does not present an analysis plan.
Given the ambition of the project, readers need an outline of:
• the growth modelling approach (e.g., multilevel modelling, latent growth curves),
• how missing data and dropout will be handled,
• how time will be conceptualized across overlapping cohorts.
Without such guidance, the statistical coherence of the project is difficult to evaluate.
Recommendation: Add a subsection with the anticipated analytical strategy for
longitudinal modelling.
3. Minor Issues1. 2. 3. 4. 5. 6. Stylistic clarity: Several sentences are long and overly dense. A stylistic pass
could improve readability without sacrificing detail.
Consistency in terminology: Terms such as “motor performance,” “physical
performance,” and “fitness” are used interchangeably. Consider standardizing
terminology.
Grammar and punctuation: Occasional small issues (e.g., missing articles,
double periods) should be corrected during final editing.
Figure descriptions: Figure 1 and Figure 2 could benefit from more detailed
captions explaining their relevance.
Motivation and grit scales: While adaptations are mentioned, clarity is needed
on whether translated/adapted items underwent any psychometric validation in
this population.
Ethics statement: Already included, but consider indicating whether assent
procedures for minors were applied.
4. Recommendation
Decision: Major Revision
The manuscript is strong and important, but it requires clearer articulation of research
aims, improved theoretical integration, and more methodological transparency before it
is fully publishable.
Reviewer 2 Report
Comments and Suggestions for Authors
OVERALL COMMENT:
The authors present methodology and framework of a novel large-cohort longitudinal individual, performance, health, and environmental characteristics on competitive outcomes in young track and field athletes.
In the introduction of the manuscript the authors provide an overview of the bioecological model used as framework for the investigation, pointing out how this is the first study to employ this methodology in young track and field athletes.
In the methods section, authors describe how each component of the bioecological model is tested. In this section some inconsistencies are present in terms of verb tenses used. Analysis of sprint running, due to the collection of numerous intermediate times, could be expanded whith analysis of the force-velocity profile (Samozino et al., 2008; Duca et al., 2020). Additionally, considerations should be made regarding the potential effects of dropout on statistical power and sample size.
In the discussion, authors describe the benefits of the study and potential limitations.
In parts of the manuscript, the first plural person “we” is chosen – authors should refrain from this choice and rephrase theses sentences.
The manuscript will benefit from a MINOR REVISION
Please refer to the attached word document for specific comments

Reviewer 3 Report
Comments and Suggestions for Authors
Firstly, I would like to thank you for the opportunity to review this article. I would like to emphasise that the comments made here are intended to improve the article, not to belittle it.
Specific comments: The article presents an opportunity to expand upon the existing literature on the subject. However, it is imperative to consider a number of variables:
Title: The title is clear and expresses conformity with the study proposal.
The Abstract is competently written and provides a lucid overview of the study's aim and proposal. However, when using keywords, try to use different words to those in the title. It is vital to ensure that the keywords do not coincide with those present in the title, as this will serve to minimise the probability of a researcher locating the paper.
The Introduction is well-written and presents a good definition of the problem and aims. However, it could be improved and some points should be considered:
It is important to maintain a clear focus on the specific gap in athletics studies and justify the need for the study.
It is recommended that the following points are emphasised more explicitly:
- why previous longitudinal studies do not meet current needs;
- how EXPERT overcomes the methodological limitations of these studies.
Line 45, please, correct the points after etc “… attributes, etc..)”
Material and Methods
It's relatively well-written, but needs some tweaking.
On item 2.1 Desing, lines 139-141, please consider deleting the sentence “As previously mentioned, the primary goal of the EXPERT study is to investigate the dynamics of the relationships between the characteristics of young athletes and multiple environmental factors that can affect their motor performance and health-related traits” since, as said, it has already been previously mentioned.
Please, try to reduce the size of Figure 1, either by removing information or by changing its formatting.
On item 2.2 Sample, the description is good, but it would be useful to include a brief explanation of the inclusion and exclusion criteria (e.g. the minimum amount of time spent participating in athletics and health-related factors).
Table 1 is clear, but you could consider:
- adding a more explanatory legend;
- providing a brief description of the rationale for age overlap.
In item 2.3. (Procedures, lines 185-187), please provide a detailed description of the 'general warm-up'. Specify the intensity of the continuous running and detail which mobility exercises and running technique drills are used.
In item 2.4.2. Performance domain, it is recommended that the reasons for including or excluding certain tests, especially those related to technical skills specific to athletics, are justified.
Discussion:
The discussion is well-founded, but:
It rehashes literature that was already cited in the introduction at length – it is suggested being more concise in some parts.
Some statements could be more critical, especially regarding the expected limitations of the mixed-longitudinal design (e.g. cohort effects).
It would be useful to:
- Reinforce the practical importance of the data for national sports policies.
- Provide examples of how coaches can use the findings in future.
Regarding limitations, although well written, it could include:
- limitations in causal inference due to the observational nature of the study;
- possible self-selection bias (structured clubs and athletes may be more likely to adhere).
Conclusions:
The conclusion it is concise and in line with the aim of the study.
References: A considerable proportion of the references are relevant to the subject under discussion.
Round 2
Reviewer 1 Report
Comments and Suggestions for Authors
MINOR REVISIONS - EXPERT Manuscript v2
OVERALL RECOMMENDATION: ACCEPT WITH MINOR
REVISIONS
The manuscript has improved substantially. Most major concerns have been addressed,
particularly the statistical analysis section. However, several minor issues require
correction.
ESSENTIAL CORRECTIONS
1. ABSTRACT - Needs restructuring
The abstract lacks clear structure. Revise to:
"Background: Longitudinal research examining individual-environment interactions in
youth athletic development is scarce for track and field. Objectives: The EXPERT study
investigates how individual (anthropometry, maturation, motivation) and environmental
(family, coach, club) characteristics influence developmental trajectories in youth track
and field athletes. Methods: A mixed-longitudinal design will follow 400 athletes (200♂,
200♀; aged 10-14 years) from 40 Portuguese clubs across five cohorts assessed
biannually over three years. Guided by Bronfenbrenner's bioecological model,
assessments encompass individual, performance, health, and environmental domains.
Multilevel growth models will examine trajectories and predictor effects. Conclusions:
EXPERT will provide evidence to optimize training and support holistic youth athlete
development."
2. KEYWORDS - Improve specificity
Replace current keywords with: "youth athletes; track and field; mixed-longitudinal
design; biological maturation; motor performance; bioecological model"
These terms improve indexation and align with international standards.
3. INTRODUCTION - Clarify previous study limitations
The description of TOYA, GYS, and INEX limitations is vague. Specify: "Although the
TOYA, GYS, and INEX studies advanced understanding of youth athlete development,
none examined track and field athletes. Moreover, these studies focused primarily on
physical growth and motor performance, with limited consideration of psychological
traits (motivation, perseverance), health indicators (nutritional status, postural control),
and environmental factors (coach behaviors, club resources)."
4. SPECIFIC AIMS - Condense
The four specific aims are too detailed for the introduction. Reduce to: "Complementary
specific aims include: (i) modeling growth dynamics via distance- and velocity-based
curves; (ii) aligning motor performance with peak height velocity to identify sensitive
periods; (iii) examining trajectory associations with individual and environmentalcharacteristics; and (iv) quantifying inter-individual variability and identifying
predictors."
5. SECTION 2.2 - Reorder subsections
Current order is illogical (recruitment before sample size justification). Restructure as:
• 2.2 Sample [general description]
• 2.2.1 Sample size calculation
• 2.2.2 Recruitment and eligibility criteria
6. EXCLUSION CRITERIA - Specify completely
Current criteria are incomplete. Revise to: "Exclusion criteria include: (i) age <10 or >14
years at baseline; (ii) physical, cognitive, or developmental disabilities requiring adapted
training; (iii) acute or chronic conditions contraindicating physical testing (per medical
documentation); and (iv) inability to provide informed consent/assent."
7. WARM-UP DESCRIPTION - Reduce detail
The warm-up description is excessively detailed. Condense to: "The general warm-up
will consist of continuous running (~5 minutes, low intensity), joint mobility exercises
(foot, hip, shoulder circles; knee flexion/extension; lunges; torso twists; leg swings),
running technique drills (arm swings, ankle drills, skipping), and short accelerations (10
m). Total duration: approximately 20 minutes."
8. MATURITY OFFSET - Acknowledge limitations
Add after the maturity offset description: "Although the maturity offset method has
recognized limitations (prediction error ≈±1 year, reduced accuracy in early/late
maturers), it remains the most feasible non-invasive approach for large-scale field studies.
Sensitivity analyses will compare results using chronological age versus maturity offset
to assess estimation error impact."
Include reference: Malina, R.M.; Rogol, A.D.; Cumming, S.P.; Coelho e Silva, M.J.;
Figueiredo, A.J. Biological maturation of youth athletes: Assessment and implications.
Br. J. Sports Med. 2015, 49, 852-859.
9. PSYCHOMETRIC VALIDATION - Add specific data
The validation paragraph mentions validation but provides no evidence. Add: "The
translated items have been psychometrically validated in Portuguese youth. The Intrinsic
Motivation Inventory demonstrated adequate internal consistency (Cronbach's α: .70-.85)
and satisfactory four-factor structure fit. The Short Grit Scale showed acceptable
reliability (perseverance: α=.78; consistency: α=.73) and model fit (CFI=.94,
RMSEA=.06), supporting construct validity in this population."
10. SPLIT TIMES - Clarify purposeExpand the excellent addition about split times: "Split times will be collected to
characterize performance across acceleration phases (0-5m: reaction/initial acceleration;
5-20m: acceleration; 20-40m: maximum velocity transition) rather than to derive
mechanical force-velocity parameters."
11. HURDLE STEP SUBSAMPLE - Justify 55%
Provide clearer rationale: "Dynamic postural control will be assessed with the Hurdle
Step test in a stratified random subsample (55%, n≈220), given time constraints. This
subsample provides adequate power (>.80) to detect medium effects (f=.25) while
reducing assessment burden. Athletes will be randomly selected within sex × cohort strata
at baseline and reassessed at all time points."
12. STATISTICAL ANALYSIS - Minor clarifications
Revise awkward phrasing:
• "handling missing data will be achieved by analyzing" → "Missing data patterns
will be analyzed to determine missingness mechanisms"
• "as well as their association with" → "and their associations with"
13. DISCUSSION - Condense substantially
The discussion is too long and repetitive with the introduction. Reduce by
approximately 50% by:
• Eliminating redundant background on developmental models
• Shortening the review of previous longitudinal studies
• Condensing the limitations paragraph
• Streamlining expected outcomes
Suggested structure: (1) brief context (2 paragraphs), (2) study strengths (1 paragraph),
(3) limitations (1 paragraph), (4) expected contributions (1 paragraph).
14. MINOR GRAMMATICAL CORRECTIONS
• "It is known" → "Adolescence is recognized as"
• "adopts the Bronfenbrenner's" → "adopts Bronfenbrenner's"
• "carry out a warm-up prior to" → "complete a warm-up before"
• "assessed perseverance" → "assesses perseverance" (tense consistency)
• "will be removed" → "was removed" (already validated)
15. FIGURE 2 CAPTION - Improve concision
Revise to: "Figure 2. Assessment sequence in the EXPERT study. Following athlete
identification and questionnaire completion, participants undergo anthropometry and
hurdle step assessment. After standardized warm-up, athletes complete motor
performance tests in the order shown: handgrip strength, 40m sprint (with splits), sit-ups,medicine ball throws, jumps (standing long jump, squat jump, countermovement jump,
reactive strength), and 1000m run."
16. SUPPLEMENTARY FILE - Verify availability
The manuscript references "supplementary file 1" for additional domain details. Ensure
this file is actually provided with submission.
17. REFERENCE FORMATTING - Standardize
Check all references for consistent formatting, particularly capitalization and journal
abbreviations, which vary throughout.
